# Targeted Imaging of Lung Cancer with Hyperpolarized ^129^Xe MRI Using Surface-Modified Iron Oxide Nanoparticles as Molecular Contrast Agents

**DOI:** 10.3390/cancers14246070

**Published:** 2022-12-09

**Authors:** Atsuomi Kimura, Seiya Utsumi, Akihiro Shimokawa, Renya Nishimori, Rie Hosoi, Neil J. Stewart, Hirohiko Imai, Hideaki Fujiwara

**Affiliations:** 1Department of Medical Physics and Engineering, Area of Medical Imaging Technology and Science, Division of Health Sciences, Graduate School of Medicine, Osaka University, Osaka 565-0871, Japan; 2POLARIS, Imaging Sciences, Department of Infection, Immunity & Cardiovascular Disease, University of Sheffield, Sheffield S10 2TA, UK; 3Division of Systems Informatics, Department of Systems Science, Graduate School of Informatics, Kyoto University, Kyoto 606-8561, Japan

**Keywords:** hyperpolarized ^129^Xe MRI, murine lung cancer, targeted imaging, molecular targeting contrast agent

## Abstract

**Simple Summary:**

Hyperpolarized ^129^Xe (HP ^129^Xe) MRI was used to demonstrate the feasibility of targeted imaging of lung cancer using two cancer-specific surface-modified iron oxide nanoparticles (IONPs) as negative contrast agents; polyethylene glycol-coated IONPs (PEG-IONPs), and dextran-coated IONPs conjugating folate on their surface (FA@Dex-IONPs). After the intravenous injection of IONPs, HP ^129^Xe signal reductions were observed at cancer sites for both PEG-IONPs and FA@Dex-IONPs administered in mice. By targeted imaging with HP ^129^Xe MRI, characteristic differences in pharmacokinetics between PEG-IONPs and FA@Dex-IONPs were successfully monitored. In particular, FA@Dex-IONPs were found to show superior pharmacokinetics for HP ^129^Xe MRI in terms of prolonged clearance due to their binding to overexpressed folate receptors in lung cancer cells.

**Abstract:**

Hyperpolarized ^129^Xe (HP ^129^Xe) MRI enables functional imaging of various lung diseases but has been scarcely applied to lung cancer imaging. The aim of this study is to investigate the feasibility of targeted imaging of lung cancer with HP ^129^Xe MRI using surface-modified iron oxide nanoparticles (IONPs) as molecular targeting contrast agents. A mouse model of lung cancer (LC) was induced in nine mice by intra-peritoneal injection of urethane. Three months after the urethane administration, the mice underwent lung imaging with HP ^129^Xe MRI at baseline (0 h). Subsequently, the LC group was divided into two sub-groups: mice administered with polyethylene glycol-coated IONPs (PEG-IONPs, *n* = 4) and folate-conjugated dextran-coated IONPs (FA@Dex-IONPs, *n* = 5). The mice were imaged at 3, 6, and 24 h after the intravenous injection of IONPs. FA@Dex-IONPs mice showed a 25% reduction in average signal intensity at cancer sites at 3 h post injection, and a 24% reduction at 24 h post injection. On the other hand, in PEG-IONPs mice, while a signal reduction of approximately 28% was observed at cancer sites at 3 to 6 h post injection, the signal intensity was unchanged from that of the baseline at 24 h. Proton MRI of LC mice (*n* = 3) was able to detect cancer five months after urethane administration, i.e., later than HP ^129^Xe MRI (3 months). Furthermore, a significant decrease in averaged ^1^H T2 values at cancer sites was observed at only 6 h post injection of FA@Dex-IONPs (*p* < 0.05). As such, the targeted delivery of IONPs to cancer tissue was successfully imaged with HP ^129^Xe MRI, and their surface modification with folate likely has a high affinity with LC, which causes overexpression of folate receptors.

## 1. Introduction

Over the past 30 years, hyperpolarized (HP) noble gas MRI, which can extraordinarily improve the MR signal of noble gases such as ^3^He and ^129^Xe, has been developed from a concept to a tool for studying lung disease ex vivo and in a preclinical setting. Recently, human clinical trials are underway [1,2]. In particular, in 2015, the UK Medicines and Healthcare Products Regulatory Agency approval was obtained at the University of Sheffield in the UK for the clinical application of HP ^129^Xe MRI (see e.g., [3]). Visualization of HP noble gas inhaled into the lungs provides unique information about lung function, such as ventilation and alveolar microstructure. To date, the technique has been applied to study respiratory pathologies such as cystic fibrosis [4], asthma [5], chronic obstructive pulmonary disease (COPD) [6,7,8], idiopathic pulmonary fibrosis [9], and others [10].

Nevertheless, the application of HP noble gas MRI to lung cancer has been limited by the lack of specific interaction between noble gases and cancer cells. Previous studies were performed only after solid cancer nodules had fully developed [11,12,13,14,15]. For the practical use of lung cancer imaging by HP noble gas MRI and prior to consideration for clinical trials, several preclinical studies using small animals such as mice and rats have been performed [16,17,18,19,20,21]. In one of these studies, targeted imaging of lung cancer was proposed by applying a negative contrast medium of magnetic nanoparticles conjugating cancer-specific targeting molecular probes on their surfaces to HP ^3^He MRI [17]. This method may hold promise for early detection of lung cancer; however, its application has been limited to date and there is a limited supply of ^3^He worldwide. To this end, we have attempted to improve the methodology to produce hyperpolarized xenon (HP ^129^Xe) and have developed a preclinical HP ^129^Xe MRI system with a custom-built flow-mode polarization apparatus to deliver it to a mouse in a closed loop [21]. Here, in order to enable lung cancer imaging with HP ^129^Xe MRI, we apply our preclinical system to targeted imaging of lung cancer using magnetic nanoparticle-based contrast media.

There are two main approaches for the targeted delivery of magnetic nanoparticles into lung cancer tissue. One is passive cancer-targeting and is based on the enhanced permeability and retention (EPR) effect [22], and the other is active targeting. Magnetic particles with a size range of 10–100 nm are used for passive targeting because they can permeate into cancer tissues that have defective endothelial cells with a wide opening. Furthermore, polyethylene glycol is usually added to the surface of the magnetic nanoparticles via so-called PEGylation to avoid phagocytosis by the reticuloendothelial system. This prolongs their retention time in blood and promotes the EPR effect. For active targeting, cancer-specific targeting molecular probes are conjugated on the surface of magnetic nanoparticles, resulting in specific uptake of magnetic nanoparticles in cancer cells and accumulation in cancer tissue. One candidate for an active molecular probe is folic acid [23]. Folic acid binds to folate receptors which are overexpressed on the surface of various cancer cells including lung cancer but expressed at low levels in normal cells.

In the present study, two types of magnetic nanoparticles are applied to targeted imaging of lung cancer by HP ^129^Xe MRI. One is PEGylated iron oxide nanoparticles (PEG-IONPs) with a hydrodynamic size of 31 nm, and the other is dextran-coated iron oxide (Fe_3_O_4_) nanoparticles conjugating folate on their surface (FA@Dex-IONPs) with a hydrodynamic size of 78 nm. In addition, characteristic pharmacokinetic behaviors of the two IONPs in lung cancer tissue are investigated by HP ^129^Xe MRI.

## 2. Materials and Methods

### 2.1. Materials

All used chemicals in the present study were in the analytical grade and used without any further purification. PEG-IONPs with a hydrodynamic size of 31 nm were purchased from Sigma-Aldrich. Iron (III) chloride hexahydrate (FeCl_3_ˑ6H_2_O), iron (II) chloride tetrahydrate (FeCl_2_ˑ4H_2_O), *N,N’*-dicyclohexylcarbodiimide (DCC), and anhydrous dimethyl sulfoxide (DMSO) were purchased from Kanto Chemical Co., Inc. (Tokyo, Japan). Sodium hydroxide (NaOH) was purchased from FUJIFILM Wako Pure Chemical Corporation (Osaka, Japan) and 4-dimethylamino pyridine (DMAP) was purchased from Kishida Chemical Co., Ltd. (Osaka, Japan). Folic acid and Dextran 40 (Mw = ca. 40 kDa) were purchased from Tokyo Chemical Industry Co., Ltd. (Tokyo, Japan).

### 2.2. Synthesis of FA@Dex-IONPs

First, dextran-coated iron oxide (Fe_3_O_4_) nanoparticles (Dex-IONPs) were synthesized according to the reported method [24]. In brief, FeCl_2_ˑ4H_2_O and FeCl_3_ˑ6H_2_O were mixed in a 1:2 molar ratio (250 mg and 665 mg, respectively) and dissolved in 80 mL of distilled water. Dextran (1.56 g) was added to the solution and the mixture was heated to 80°C. After purging the solution with nitrogen for 1 hour, 1.0 M NaOH solution (12 mL) was added dropwise to the mixture. The reaction was carried out for 1 hour. The resulting black suspension was magnetically decanted and washed three times with deionized water. The Dex-IONPs were dissolved in 5 mL of deionized water and probe sonicated for 10 min. The Dex-IONPs were separated from unbound dextran by gel filtration chromatography using Sepharose gel in a CL-2B column (GE Healthcare Japan Co., Ltd., Tokyo, Japan). The Dex-IONPs were dialyzed against deionized water for 48 h (12 kDa molecular weight cutoff) to neutralize the solution.

Subsequently, the surface of the Dex-IONPs was conjugated with folic acid according to the reported method [25]. In brief, DCC (0.2 g), DMAP (0.1 g), and folic acid (0.4 g) were dissolved in anhydrous DMSO (80 mL) and stirred under nitrogen atmosphere at room temperature overnight. Then 20 mL of the Dex-IONPs aqueous solution (5 mg/mL) was added to the reaction mixture and stirred for 24 h at 80°C under a nitrogen atmosphere and in darkness. The FA@Dex-IONPs were magnetically decanted and washed with ethanol and water three times. Finally, the FA@Dex-IONPs were suspended in the deionized water (4 mg/mL) and probe sonicated for 10 min.

The hydrodynamic size of the FA@Dex-IONPs was measured to be 78 nm by Zetasizer Pro (Malvern Panalytical Ltd., Malvern, UK). The folate content in FA@Dex-IONPs was quantified from the absorbance at 363 nm, in which a calibration curve was developed by measuring the intensity of 363 nm absorbance using a spectrophotometer, ASV 11D (As One corporation, Osaka, Japan), as a function of folate concentration. The folate content of the resulting FA@Dex-IONPs was estimated to be 0.2 mM/mg/mL using the calibration curve.

### 2.3. Measurement of Net Iron Concentration in IONPs

The net iron concentration in IONPs was determined according to a reported method [26]. First, 0.225 mL of concentrated HCl (37%) was added to 25 μL of IONP solution to ionize the iron oxide crystal core and to liberate the iron in its ferric state. An amount of 0.250 mL of a 40 mM potassium thiocyanate solution was then added to the sample. Then, the sample was diluted 6-fold. The absorbance at a wavelength of 480 nm was read using a spectrophotometer, ASV 11D. An aqueous solution of Fe_3_O_4_ was used to record the calibration curve.

### 2.4. Animal Preparation

All experimental and animal care procedures conformed to the guidelines of Osaka University.

Nine male, six-week-old ddY mice (Japan SLC, Inc., Shizuoka, Japan) were included in this study. A 500 μL saline solution of urethane (Tokyo Kasei, Tokyo, Japan) was intra-peritoneally administrated to each mouse to induce lung cancer (500 mg/kg body weight). Similarly, a negative-control (NC) group (*n* = 8) was prepared by intra-peritoneal administration with a 500 μL saline solution. Three months after urethane/saline injection, the mice underwent non-invasive respiratory-gated HP ^129^Xe MRI as described below. Mice initially underwent one lung HP ^129^Xe MRI exam at baseline, i.e., preceding administration of IONPs (0 h). Subsequently, the lung cancer (LC) mice were further divided into two subgroups: mice administered with PEG-IONPs (PEG-IONPs LC group, *n* = 4) and FA@Dex-IONPs (FA@Dex-IONPs LC group, *n* = 5). The NC mice were also divided into two subgroups: mice administered with PEG-IONPs (PEG-IONPs NC group, *n* = 4) and FA@Dex-IONPs (FA@Dex-IONPs NC group, *n* = 4). The IONPs solution was intravenously administered to the LC and NC mice at a dose of 61 μmol Fe/kg into the tail vein. HP ^129^Xe MRI was performed for the mice at 3, 6, and 24 h after the IONPs administration. The survival rate of the whole procedure was 100% for all groups.

### 2.5. HP ^129^Xe MRI

All MR measurements were performed on an Agilent Unity INOVA 400WB (Agilent Technologies, Inc., Santa Clara, CA, USA) with a 9.4 T vertical magnet (Oxford Instruments Plc., Oxford, UK), and a Highland L-500 Gradient Amp system (Highland Technology, Inc. SFO, CA, USA). A self-shielded imaging probe (Litz coil switchable between ^129^Xe and ^1^H frequencies) with a 32-mm diameter and a 15-mm length (Clear Bore DSI-1117, Doty Scientific, Inc., Columbia, SC, USA) was used.

Immediately before all MR measurements, mice were anesthetized with 2% isoflurane. HP ^129^Xe was produced with a home-built continuous-flow ^129^Xe polarizer as previously described [21]. A gas mixture of HP ^129^Xe and N_2_ (^129^Xe in natural abundance (^129^Xe fraction 0.26): 70 %, N_2_: 30 %) was continuously delivered to the mouse in the MRI scanner at a flow rate of 50 mL/min after the gas mixture was mixed with O_2_ (flow rate 12 mL/min). The mice spontaneously breathed the polarized gas through a mask attached to the head. Respiratory-gated HP ^129^Xe MRI measurements were performed using a balanced steady-state free precession (bSSFP) sequence with two saturation (90°) pre-pulses in order to destroy the HP magnetization. After the HP magnetization was destroyed, HP ^129^Xe coronal images of the lungs were acquired at an inspiratory phase over 10 breathing cycles. A respiratory sensor was used to synchronize the acquisition of HP ^129^Xe images to the respiratory signal as described previously [21]. Throughout the MRI process, the body temperature of the mice was maintained with warm water circulating through a rubber tube placed on the abdomen. Thus, HP ^129^Xe MRI was performed without tracheal intubation or tracheotomy and was hence entirely noninvasive.

Acquisition parameters of HP ^129^Xe images were as follows: 1000-μs gaussian-shaped RF pulses of flip angle α = 180° with a preparation pulse of α/2; acquisition bandwidth, 88 kHz; TR/TE = 3.6ms/1.8ms; echo train length, eight; number of shots, four; number of averages, 24; coronal slice thickness, 20 mm; matrix, 64 × 32 (reconstructed to 128 × 64) with a field of view of 80 × 25 mm^2^.

### 2.6. Proton MRI

Proton MRI was performed 5 months after the urethane administration. After proton MRI was measured at baseline (0 h), FA@Dex-IONPs were administered to mice at a dose of 61 μmol Fe/kg. Subsequently, the mice were imaged at 6 and 24 h post administration of FA@Dex-IONPs. A multi-echo multi-slice sequence was used to acquire images with parameters as follows: TR = 4000 ms, TE = 14, 21, 28, 35 ms; number of excitations, four; scan time, 34 min; slice thickness = 0.7 mm; FOV = 3 × 3 cm; slices = 8; matrix = 128 × 128. During scanning, mice were anesthetized with 2% isoflurane. Images were analyzed to obtain T2 maps using Image J.

### 2.7. Lung Cancer Site Counting

After HP ^129^Xe MRI, LC mice were euthanized with a lethal dose of carbon dioxide gas to measure the number of lung surface cancer sites. The lungs were extracted and immersed in 10% formalin at 25 cmH_2_O for at least 1 week. The location of surface cancers on the lung was confirmed visually.

### 2.8. Image Analysis

The ROIs of HP ^129^Xe coronal image of lung from each LC mouse were divided into two groups; the ROI containing lung cancer (ROI _LC_cancer_) and the ROI with no established cancer (ROI _LC_no cancer_). The ROI _LC_cancer_ was set using ImageJ (National Institutes of Health) as illustrated in Figure 1. First, the difference image was created between the images acquired at 0 h and each time point post IONPs administration after registering the image at each time point with the image at 0 h. Then, the difference image was used to depict the edge of the ROI _LC_cancer_ using a threshold determined by Otsu’s method [27] and by visually referring to the locations of cancers determined in the previous section. The ROI _LC_no cancer_ was depicted by subtracting the ROI _LC_cancer_ from whole lung image.

The time course analysis of images was made for each ROI of cancer and no cancer. First, the mean signal intensity was derived within a ROI at time t to give *Mean(t, ROI)*, and divided by the maximum signal intensity for the whole lung, *Max(t, whole lung)*, to give the ratio *SI(t, ROI)*:(1)$$SI\left(t,\;ROI\right)=\frac{Mean\left(t,\;ROI\right)}{Max\left(t,\;whole\;lung\right)}\;t\;=\;0,\;3,\;6,\;24\;(h)$$

Here, the ratio was taken to compensate for the total signal fluctuations in the time course measurement. Second, time course variation was represented by the relative value to the initial value at *t* = 0:(2)$$Relative\;SI\left(t,\;ROI\right)=\frac{\;SI\left(t,\;ROI\right)}{\;SI\left(0,\;ROI\right)}$$

Similarly, the relative signal intensity of the whole lung of each NC mouse (ROI _NC_) was also evaluated.

### 2.9. Histology

After HP ^129^Xe MRI measurements were made and lung cancer was counted, the lungs of the LC model mice were processed for histology by Prussian Blue nuclear fast red staining. Six coronal whole lung slices were obtained from each mouse and then captured using a microscope (Wraymer BX-2700 TL; Wraymer Inc., Osaka, Japan).

### 2.10. Statistical Analysis

Statistical analysis was performed by one-way ANOVA with a Tukey–Kramer post-hoc analysis to identify significant differences between the LC and NC model groups. All data are presented as mean ± standard error and/or box-and-whisker plots. Differences in signal intensities are considered significant at the *p* < 0.05 level.

## 3. Results

Figure 2 shows the results of targeted imaging of the lung cancer with HP ^129^Xe MRI for LC mice. Figure 2a,b demonstrates the time-course of HP ^129^Xe images acquired before and after FA@Dex-IONPs and PEG-IONPs administration, respectively, compared with HP ^129^Xe images for NC mice before and after IONPs administration.

Comparison of HP ^129^Xe images with photographs of the lungs confirmed that signal voids were observed at the ROIs corresponding to the lung cancer locations at some time points for both the FA@Dex-IONPs LC and PEG-IONPs LC groups. On the other hand, high signal intensity with a homogeneous distribution across the lungs was preserved for all time points for both the FA@Dex-IONPs NC and PEG-IONPs NC groups. Since signal voids were not observed at any time point for the NC mice, the signal voids observed for the LC mice can be attributed to the presence of the IONPs.

Figure 3 shows the time-course of the HP ^129^Xe relative signal intensities obtained using Equation (2) for LC and NC mice. The averaged relative signal intensities from ROIs containing cancer of LC mice (ROIs _LC_cancer_) were compared with those from ROIs containing no cancer, for LC mice (ROIs _LC_no cancer_) and NC mice (ROIs _NC_).

For FA@Dex-IONPs LC mice, the averaged relative signal intensities from ROIs _LC_cancer_ were significantly lower at 3, 6, and 24 h post injection than those from ROIs _LC_no cancer_ (*p* < 0.01) and ROIs _NC_ (*p* < 0.01). Meanwhile, for PEG-IONPs LC mice, the averaged relative signal intensities from ROIs _LC_cancer_ were significantly lower only at 3 and 6 h post injection than those from ROIs _LC_no cancer_ (*p* < 0.01 at 3 h and *p* < 0.05 at 6 h, respectively) and ROIs _NC_ (*p* < 0.01).

Figure 4 shows a box-and-whisker plot comparison of the averaged relative signal intensities of FA@Dex-IONPs LC mice 24 h after administration with those of PEG-IONPs LC, FA@Dex-IONPs NC, and PEG-IONPS NC mice 24 h after administration. The averaged relative signal intensity of FA@Dex-IONPs LC mice was significantly lower than that of PEG-IONPs LC mice (*p* < 0.05). From Figure 3 and Figure 4, it is clear that there is a characteristic difference between the pharmacokinetic action of FA@Dex-IONPs and PEG-IONPs.

Figure 5 shows the representative Prussian Blue nuclear fast red-stained histology images of LC mice 24 h after FA@Dex-IONPs and PEG-IONPs administration. The presence of iron in the cancer tissue was visually confirmed on histology images of FA@Dex-IONPs LC mice (Figure 5a) whereas it was difficult to observe the accumulation of PEG-IONPs in lung cancer (Figure 5b).

Figure 6 shows the results of proton MRI of LC mice administered with FA@Dex-IONPs. Proton MRI was able to detect cancer 5 months after urethane administration. Figure 6a demonstrates the T2 map at lung cancer sites superimposed on the proton image acquired at TE = 14 ms. Figure 6b shows the temporal change of T2 values at cancer sites. Although there was a significant difference between T2 values at baseline (0 h) and 6 h post administration of FA@Dex-IONPs (*p* < 0.05), no significant difference was observed between T2 values assessed at 0 h and 24 h.

## 4. Discussion

In the present study, the use of IONPs as a negative contrast agent has enabled targeted imaging of lung cancer with HP ^129^Xe MRI (Figure 2). Furthermore, by comparing two contrast agents of FA@Dex-IONPs and PEG-IONPs, it was observed that there is a characteristic difference in their pharmacokinetics (Figure 3). In particular, FA@Dex-IONPs were found to exhibit superior pharmacokinetic properties compared with PEG-IONPs (Figure 3 and Figure 4). Aside from one report of targeted imaging of lung cancer with HP ^3^He [17] and a few reports of cancer MRI using folate-modified IONPs [25,28], to our knowledge, this is the first application of HP ^129^Xe to lung cancer imaging using IONPs.

The contrast enhancement effect of FA@Dex-IONPs was also examined by proton MRI. A significant decrease in T2 values at cancer sites was observed at 6 h post FA@Dex-IONPs administration, but no significant difference was observed between baseline (0 h) and 24 h. (Figure 6). This means the contrast enhancement effect of FA@Dex-IONPs at a dose of 61 μmol/kg was insufficient and/or too temporally short-lived for targeted imaging with proton MRI, indicating the superior sensitivity of HP ^129^Xe MRI.

HP ^129^Xe MRI was performed using a bSSFP sequence because of its contrast properties [29]. Since the flip angle used in the bSSFP sequence was 180° (90° as a preparation pulse), the images have contrast similar to that obtained by using FSE sequences (Figure 2) [30]. As a result, T2-weighted images were obtained and signal reduction due to the accumulation of the negative contrast agent in the cancer tissue was observed. A previous study using HP ^3^He reported that the signal loss due to IONPs is TE-dependent, and that longer TE enhances the signal loss [17]. In the present study, based on previous reports [31,32], TE was set at 1.8 ms to produce maximum T2-weighted contrast at 9.4T. Further investigations about the exact TE dependence on contrast for each proposed IONP-based contrast agent would help to maximize the sensitivity of the technique.

In the present study, HP ^129^Xe MRI was able to detect lung cancer with a minimum diameter of about 1.1 mm. On the other hand, in proton MRI, the minimum diameter of lung cancer detected was 2.1 mm (Figure 6a), which is accordance with previously reported detectability [33]. Therefore, the sensitivity of HP ^129^Xe MRI is comparable to results attained by X-ray microcomputed tomography [34] and superior to that of proton MRI. The IONPs accumulated in cancer tissue form strong local magnetic susceptibility gradients that rapidly dephase nearby transverse magnetization. Due to the strength of this effect, a small number of IONPs can cause rapid dephasing and a significant resultant signal loss. This, in turn, boosts the sensitivity of HP ^129^Xe MRI for cancer detection. Xe spins can easily exchange between the gas phase and tissue. Hence, hyperpolarized ^129^Xe can penetrate tissue from the gas phase and sensitively detect such superparamagnetic dephasing effects before recovering to the gas phase. Such processes contribute to additional signal reduction in the alveolar gas space ^129^Xe. Xe gas possesses higher affinity to blood or tissue compared with He, as exemplified by the Ostwald solubility coefficient to blood; 0.146 for Xe and 0.010 for He [35]. Therefore, as a tool for detecting the superparamagnetic dephasing effects, ^129^Xe MRI is expected to be more sensitive than ^3^He MRI. A previous study using HP ^3^He reported that cancer lesions smaller than 300 μm can be detected by exploiting the blooming effect [17]. Therefore, optimization of gradient echo pulse sequences to effectively exploit the blooming effect warrants exploration in the attempt to further improve the detectability of small cancers.

In this study, lung cancer was induced in mice by administration of urethane. This mouse model is a multi-stage carcinogenesis model, and we previously reported that the pathological progression was characterized by epithelial hyperplasia at 1 month post urethane administration, atypical adenomatous hyperplasia at 2 months, and adenoma (solid nodule development) after 3 months [21]. As is apparent from Figure 2, lung cancer nodules could be detected as early as 3 months post urethane administration by using IONPs as a negative contrast agent in HP ^129^Xe MRI. In our previous work, we detected lung cancer 5 months post urethane administration by HP ^129^Xe MRI without IONPs [21]. Similarly, in this study, proton MRI was able to detect lung cancer 5 months post urethane administration (Figure 6a). A previous study also reported that proton contrast-enhanced MRI was able to detect lung cancers 20 weeks after urethane administration [36]. Therefore, using HP ^129^Xe MRI with IONPs in the present study, it was possible to detect lung cancer at an earlier stage.

We observed a characteristic difference in pharmacokinetic behavior of FA@Dex-IONPs and PEG-IONPs (Figure 2 and Figure 3) for imaging of lung cancer with HP ^129^Xe MRI. That is, it was confirmed that FA@Dex-IONPs accumulated in cancer tissue from 3 to 24 h after administration (Figure 2a and Figure 3a). On the other hand, PEG-IONPs accumulated in the cancer tissue between 3 and 6 h after administration, but there was no significant decrease in signal intensity 24 h after administration (Figure 2b and Figure 3b). Furthermore, the relative signal intensity at 24 h after administration achieved with FA@Dex-IONPs were significantly lower than that achieved with PEG-IONPs (Figure 4). These facts support the superiority of FA@Dex-IONPs as cancer imaging contrast agents.

There are two main possible factors for the difference in pharmacokinetic behavior between FA@Dex-IONPs and PEG-IONPs. One is the size difference; it is known that the size of nanoparticles affects their pharmacokinetics. FA@Dex-IONPs and PEG-IONPs have hydrodynamic sizes of 78 nm and 31 nm, respectively, i.e., FA@Dex-IONPs is significantly larger. Since nanoparticles with a smaller size can avoid entrapment by the reticuloendothelial system, they can circulate in the blood for a long time [37], i.e., it is anticipated that accumulation in cancer cells would continue for a long time. However, in this study, FA@Dex-IONPs with larger sizes appeared to show slower clearance. This result that cannot be explained by blood retention characteristics.

The second factor is the mechanism of cancer tissue accumulation. As mentioned in the Introduction section, the cancer accumulation mechanism of FA@Dex-IONPs is active targeting. After FA binds to folate receptors overexpressed in lung cancer, it is taken up into cells by endocytosis and trapped in cancer tissue for long periods of time [23]. We infer that this mechanism explains why the FA@Dex-IONPs continued to show contrast 24 h after administration. Histopathological examination also confirmed that FA@Dex-IONPs accumulated in cancer tissue 24 h after administration (Figure 5), supporting this hypothesis. It is also supported by reports that folate-modified drugs and nanoparticles accumulate in lung cancer in a urethane-induced lung cancer model [38,39,40,41]. On the other hand, there are a few reports that IONPs cause signal reduction at cancer tissue sites 2–3 h after administration based on the EPR effect, after which the signal reduction was gradually weakened [28,42,43,44]. This may help explain why PEG-IONPs did not exhibit a significant signal reduction 24 h after administration in the present study.

Another factor that can affect pharmacokinetic behavior is the surface charge of IONPs. Both Dex and PEG coated IONPs are reported to be negatively charged at about −10 mV [44,45]. Folate-modified IONPs are also reported to exhibit a negative surface charge of −0.2 mV [28]. The negative charge reduces affinity with cancer cells and might affect the pharmacokinetics of FA@Dex-IONPs. However, it has been reported that folate-modified IONPs were taken up by human non-small cell lung cancer cells [46]. Furthermore, it has also been reported that folate-modified nanoparticles accumulated in cancer tissue 24 h after administration despite their negative surface charge [28,39,47]. Therefore, in the present study, FA@Dex-IONPs were likely taken up by cancer cells, resulting in the improved pharmacokinetic behavior.

An advantage of the use of IONPs is that they may be applied to lung cancer treatment [25]. However, this is beyond the scope of the present study. One example is the use of magnetic hyperthermia (MHT) after IONPs have accumulated at cancer sites. There is also a report that iron oxide nanoparticles themselves inhibit tumor growth [48]. HP ^129^Xe MRI, in combination with MHT, may be highly effective in not only diagnostics, but possibly also therapy applications in the future.

A limitation of the present study is the choice of ROIs and their coarseness, which means that the locations of lung cancer sites are only roughly confirmed. In the future, pixelwise quantitative information about IONPs accumulation at cancer sites may be obtained through acquisition of a T2* map [49]. Although this is challenging in spontaneously breathing mice at present, it may be made possible by devising a ventilator and improving the resolution of respiration-gated imaging [19].

Another limitation is the type of cancer to which the present method is applicable. If cancers grow in the airway at the beginning, the ventilation image will show signal voids directly, even without the use of IONPs. Therefore, the method proposed in the present study will be effective for cancers developing in the peripheral lung. However, it should be noted that airway obstruction rarely occurs in a mouse model of lung cancer induced by urethane administration at a dose of 500 mg/kg body weight [21].

## 5. Conclusions

Lung cancer imaging with HP ^129^Xe MRI was demonstrated with the use of two cancer-specific negative contrast agents prepared based on iron oxide nanoparticles (IONPs): (i) folic acid-conjugated dextran-coated IONPs (FA@Dex-IONPs) and (ii) PEGylated IONPs (PEG-IONPs). By targeted imaging of lung cancer with HP ^129^Xe MRI, characteristic differences in pharmacokinetics between FA@Dex-IONPs and PEG-IONPs were successfully monitored. In particular, FA@Dex-IONPs were found to show superior pharmacokinetics for HP ^129^Xe MRI in terms of prolonged clearance, since they bind to folate receptors overexpressed in lung cancer cells. These findings may be useful, not only for diagnostics, but also therapy of lung cancer in the future.

## Figures and Tables

**Figure 1 cancers-14-06070-f001:**
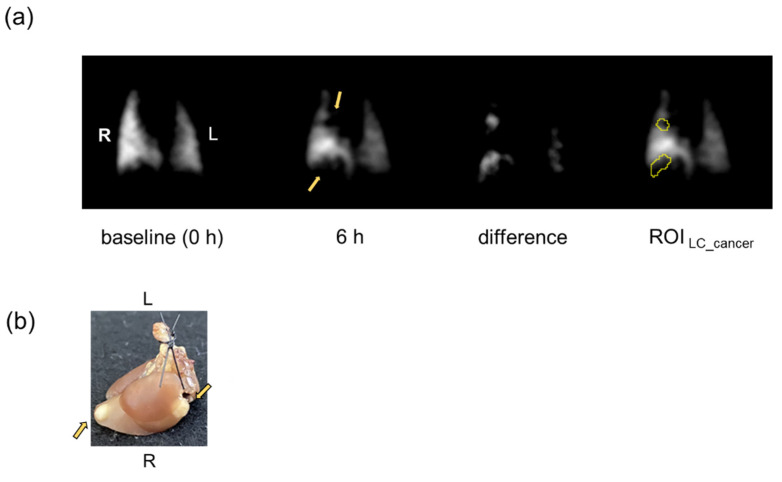
Example of ROI setting on HP ^129^Xe lung image. The ROI _LC_cancer_ was depicted using the difference image between the baseline image (0 h) and the image acquired at each time point (3, 6, and 24 h) post administration of IONPs; e.g., image 6 h post administration of PEG-IONPs shown. The arrows indicate locations of lung cancer which were qualitatively determined from comparison of the image (**a**) with photographs (**b**) of excised lung. R, right lung; L, left lung.

**Figure 2 cancers-14-06070-f002:**
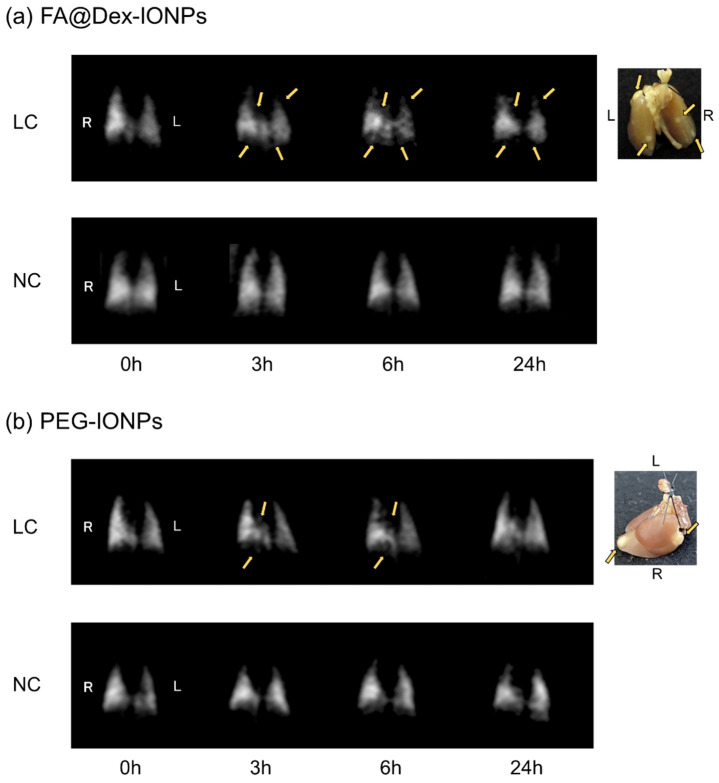
Representative examples of temporal changes of HP ^129^Xe MRI lung images pre and post FA@Dex-IONPs and (**a**) PEG-IONPs (**b**) administration together with photographs of the lung. Arrows indicate the locations of lung cancer qualitatively determined from comparison of the images with photographs of lung. Note that the PEG-IONPs LC mouse is the same as that in Figure 1. R, right lung; L, left lung.

**Figure 3 cancers-14-06070-f003:**
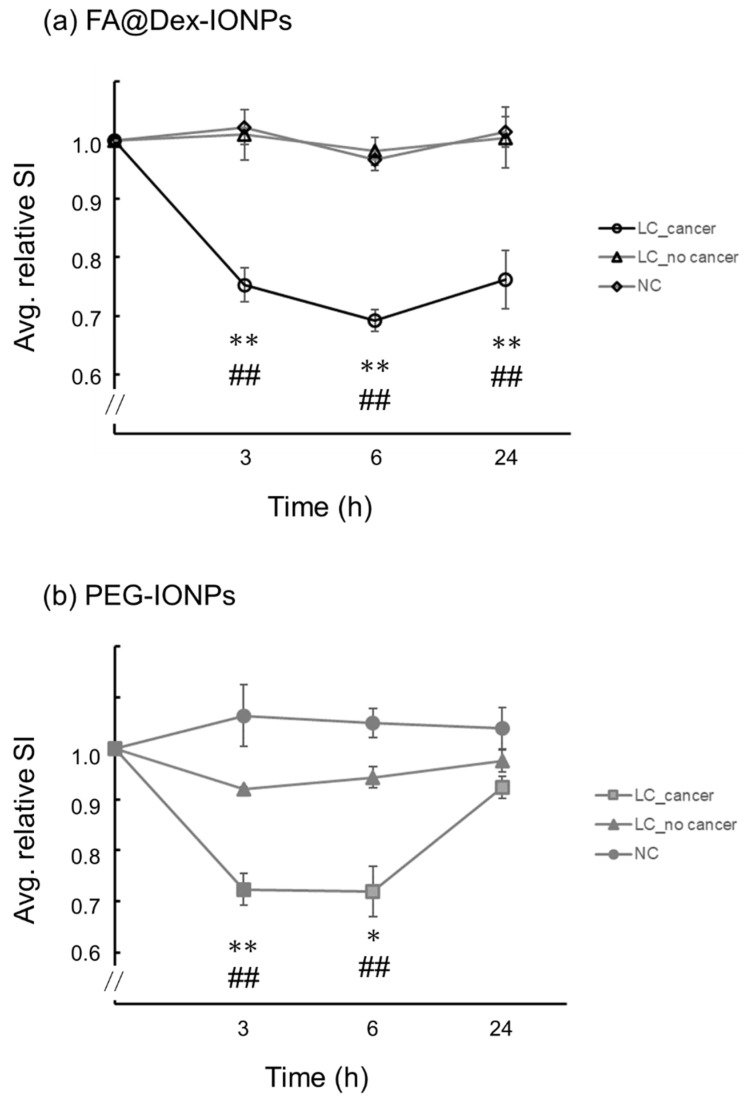
Plots of the temporal change of averaged relative signal intensity (Avg. relative SI) pre and post FA@Dex-IONPs (**a**) and PEG-IONPs (**b**) administration. The relative signal intensities were averaged for all ROIs _LC_cancer_, ROIs _LC_no cancer_, and ROIs _NC_. Data are represented by the mean ± SE. *: *p* < 0.05 between ROIs _LC_cancer_ and ROIs _LC_no cancer_, **: *p* < 0.01 between ROIs _LC_cancer_ and ROIs _LC_no cancer_, and ##: *p* < 0.01 between ROIs _LC_cancer_ and ROIs _NC_.

**Figure 4 cancers-14-06070-f004:**
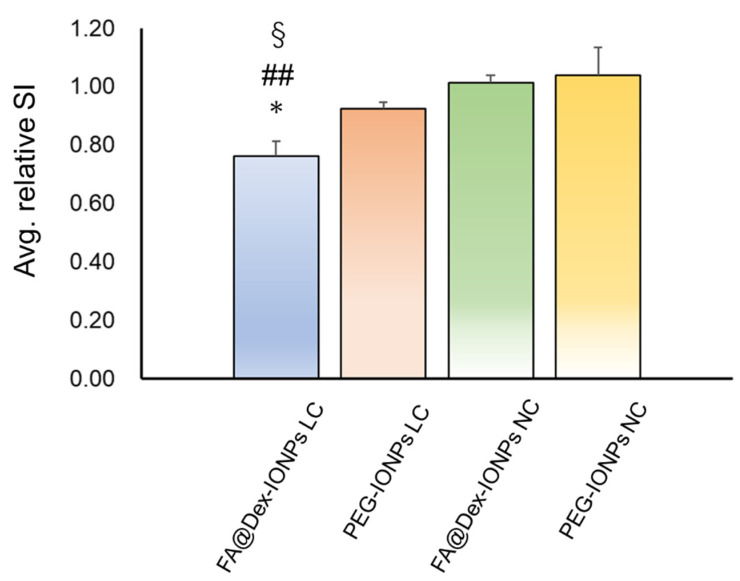
Comparison of the averaged relative signal intensity (avg. relative SI) of FA@Dex-IONPs LC, PEG-IONPs LC, FA@Dex-IONPs NC, and PEG-IONPs NC mice 24 h after administration, which were evaluated for ROIs _LC_cancer_ and ROIs _NC_. Significant differences between groups are indicated by symbols (*: *p* < 0.05 between FA@Dex-IONPs LC and PEG-IONPs LC, ##: *p* < 0.01 between FA@Dex-IONPs LC and FA@Dex-IONPs NC, §: *p* < 0.01 between FA@Dex-IONPs LC and PEG-IONPs NC).

**Figure 5 cancers-14-06070-f005:**
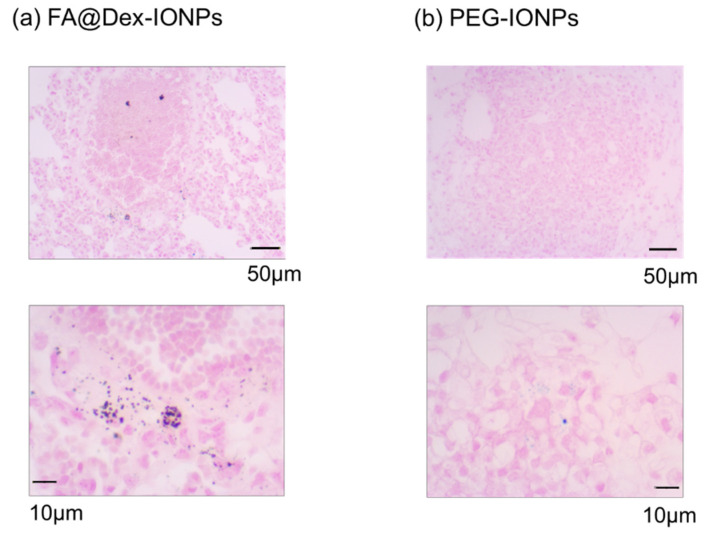
Representative examples of microphotographs of Prussian Blue nuclear fast red staining histology slides obtained from the FA@Dex-IONPs LC (**a**) and PEG-IONPs LC mice (**b**).

**Figure 6 cancers-14-06070-f006:**
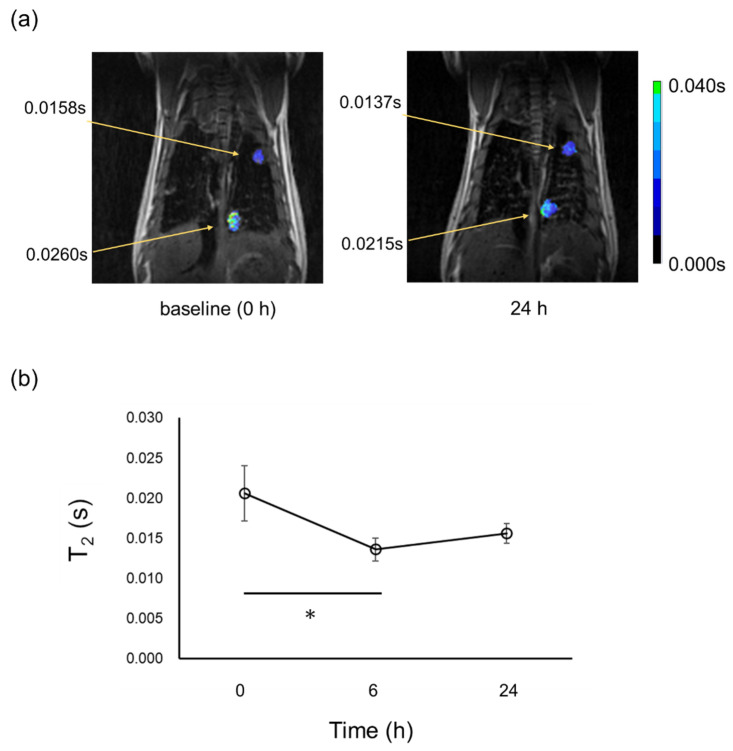
Representative T2 map at cancer sites superimposed on proton MRI of one LC mouse acquired at baseline (0 h) and 24 h after FA@Dex-IONPs administration (**a**), and temporal change of T2 values at cancer sites (**b**). T2 values at cancer sites are also included in (**a**). Significant differences between two time points are indicated by solid line (*: *p* < 0.05).

## Data Availability

Not applicable.

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
