# Peer review of "Targeted Imaging of Lung Cancer with Hyperpolarized 129Xe MRI Using Surface-Modified Iron Oxide Nanoparticles as Molecular Contrast Agents"

_cancers, 2022, doi:10.3390/cancers14246070_

Round 1

Reviewer 1 Report

In this manuscript, iron oxide nanoparticles were used as probes to enrich to lung tumors through the passive EPR effect and active binding of folic acid receptor. Iron oxide nanoparticles have paramagnetism, which can be used as relaxation agents to reduce the hyperpolarized 129Xe magnetic resonance signal in the probe rich area of the mouse lung ventilation image, so that the tumor area presents a shadow, so as to detect the tumor.

The idea of this article is similar to that of the PNAS in 2010 (PNAS, 2010, vol.107, 3693-3697). The main difference is that the article uses 3He, the mouse is passive breathing, and this article uses 129Xe, the mouse is active breathing. Another difference is that the surface layer of iron oxide nanoparticles in this article is modified with folic acid, which increases the active binding of tumor folic acid receptor, compared with iron oxide nanoparticles without modified folic acid.

For this article, I have two questions:

1. As a paramagnetic contrast agent, has the authors performed H MRI on the lungs of lung cancer mice to compare the image differences of tumor sites before and after administration of iron oxide nanoparticles?

2. From the anatomical photos of the lungs in this article, the lung tumors of mice used for imaging have formed multiple and large solid tumors. It seems that these solid tumors are located on the surface of the lung, and do not block the airway to form pulmonary ventilation obstacles. If these tumors grow in the airway at the beginning, they will show a dark area directly when scanning the ventilation image. Has this ever happened in the in vivo experiment that the authors tried? If so, can the injection of iron oxide nanoparticles make the detection of tumors earlier?

Reviewer 2 Report

The manuscript has been well written. However, few comments have been performed prior to acceptance.

1)      Manuscript would be more interesting if you have a graphical abstract.

2)      What is the difference between a simple summary and an abstract? I believe these two have to be merged.

3)      The abstract does not have quantities of results. For example: “ HP Xe signal reduction at cancer sites was observed.” How much the signal has been reduced? Etc.

4)      Quality of Figs 1(a) and 3h are very poor. (0h) and 3h cannot be distinguished by eye.

5)      Fig 4 is not informative at all. It should be presented in other way like column. 

Reviewer 3 Report

In this study Kimura et al. studied the use of two iron oxide nanoparticles, one PEGylated and the other functionalized with folate, as targeted contrast agents in hyperpolarized 129Xe MRI for murine lung cancer. The authors performed hyperpolarized 129Xe MRI using these two contrast agents and found that while both nanoparticles can result in signal reduction or contrast in tumor lesions, the folate-coated particle has longer lasting effect presumably due to better retention. While the data presented here supports the overall conclusion that using targeting IONPs can improve contrast, more data and discussions are needed to demonstrate the advantages of this approach over other existing methods. I would recommend the publication of this manuscript after the following concerns are addressed:

1.       What are the zeta potentials of the two particles? The folic acid coated particle should have a negative potential, and this can affect the uptake and distribution of the particles.

2.       What is the mass percentage of Fe for the particles? What is the actual amount of Fe injected per mice?

3.       In figure 3a, have the authors try even longer time points to see how long con the contrast effect last for FA@Dex-IONPs?

4.       In figure 3b, what is the explanation for the initial drop in signal for LC_no cancer group?

5.       In figure 5, have the authors try a more quantitative method such as ICP-MS to quantitatively compare the uptake of the two nanoparticles?

6.       There are plethora of reports using targeting iron oxide nanoparticles for MRI imaging of tumors. Have the authors try a normal (without hyperpolarized 129Xe) MRI using the two particles and compare the two methods in terms of their resolution, detection limit, etc. This could be extremely important data showing the advantage of this method.

Round 2

Reviewer 1 Report

The authors have supplemented the experimental data and answered my question. I think this manuscript can be published. However, the application scope of this method should be supplemented in the manuscript, that is, it is applicable to the detection of lung cancer types that do not block the airway.

Reviewer 2 Report

The quality of the manuscript has been improved and I recommend to accept. 

Reviewer 3 Report

The authors have addressed my previous comments to me satisfaction. 
